# Expression of pH-Sensitive TRPC4 in Common Skin Tumors

**DOI:** 10.3390/ijms24021037

**Published:** 2023-01-05

**Authors:** Bernadett Kurz, Hannah Philine Michael, Antonia Förch, Susanne Wallner, Florian Zeman, Sonja-Maria Decking, Ines Ugele, Constantin Hintschich, Frank Haubner, Tobias Ettl, Kathrin Renner, Christoph Brochhausen, Stephan Schreml

**Affiliations:** 1Department of Dermatology, University Medical Center Regensburg, Franz-Josef-Strauß-Allee 11, 93053 Regensburg, Germany; 2Center for Clinical Studies, University Medical Center Regensburg, Franz-Josef-Strauß-Allee 11, 93053 Regensburg, Germany; 3Department of Otorhinolaryngology, University Medical Center Regensburg, Franz-Josef-Strauß-Allee 11, 93053 Regensburg, Germany; 4Department of Otorhinolaryngology, University Hospital, Ludwig Maximilians University Munich, Marchioninistraße 15, 81377 Munich, Germany; 5Department of Maxillofacial Surgery, University Medical Center Regensburg, Franz-Josef-Strauß-Allee 11, 93053 Regensburg, Germany; 6Institute of Pathology, University of Regensburg, Franz-Josef-Strauß-Allee 11, 93053 Regensburg, Germany

**Keywords:** TRPC4, skin tumors, melanoma, squamous cell carcinoma, basal cell carcinoma

## Abstract

TRPCs (transient receptor potential classical or cation channels) play a crucial role in tumor biology, especially in the Ca^2+^ homeostasis in cancer cells. TRPC4 is a pH-sensitive member of this family of proteins. As solid tumors exhibit an inversed pH-gradient with lowered extracellular and increased intracellular pH, both contributing to tumor progression, TRPC4 might be a signaling molecule in the altered tumor microenvironment. This is the first study to investigate the expression profiles of TRPC4 in common skin cancers such as basal cell carcinoma (BCC), squamous cell carcinoma (SCC), malignant melanoma (MM) and nevus cell nevi (NCN). We found that all SCCs, NCNs, and MMs show positive TRPC4-expression, while BCCs do only in about half of the analyzed samples. These data render TRPC4 an immunohistochemical marker to distinguish SCC and BCC, and this also gives rise to future studies investigating the role of TRPC4 in tumor progression, and especially metastasis as BCCs very rarely spread and are mostly negative for TRPC4.

## 1. Introduction

Non-melanoma skin cancers (NMSCs) and malignant melanoma (MM) are among the most prevalent cancers in the population. In 2022, for instance, it is estimated that about 99.780 people in the US will be newly diagnosed with MM [1]. The incidence of skin cancer will increase further in the coming decades, due to the ageing population [2]. Due to its risk of metastasis, MM is responsible for 90% of deaths among skin cancers, while the group of NMSC includes mainly basal cell carcinomas (BCCs) (80%) and squamous cell carcinomas (SCCs) (20%), which rarely metastasize [3]. Even if the mortality rate and metastatic potential of NMSCs are low, these tumors lead to extensive costs for healthcare systems. Therefore, it is important to find new therapeutic targets in MM and NMSC for future treatments.

Tumor formation changes the physical microenvironment in the tissue [4]. Under physiological conditions, the extracellular pH (pH_e_) is higher (7.2–7.4) than the intracellular pH (pH_i_) (6.9–7.2), whereas in the tumor micromilieu, the so-called inverse pH gradient (pH_e_ < pH_i_) develops [5,6]. Acidic metabolic end products in the tumor microenvironment (TME) result from poor blood flow with subsequent hypoxia, as well as inflammation and high metabolic activity [6,7]. The regulation of intracellular and extracellular pH (pH_i_ and pH_e_) in tumors also depends on several transporters, such as various carbonic anhydrases (CAII, CAIX, CAXII), the sodium/hydrogen antiporter NHE1 (SoLute Carrier 9A1 [SLCA9A1]), vacuolar ATPases (V-ATPases) and many more [8]. 

The reverse pH gradient is harmful for normal cells, as cellular acidification generally leads to apoptosis. In tumor cells, on the other hand, the inverse pH-gradient leads to proliferation, evasion of apoptosis, metabolic adaption, migration and invasion, thus promoting tumor growth [9]. Unlike normal cells, tumor cells developed various mechanisms to cope with acidic and hypoxic stress by the expression of ion transporters to preserve a slightly alkaline intracellular pH or by the overexpression of proteins involved in the glucose metabolism [10]. Various methods for pH-imaging in tumors have been developed over the years, some even for in vivo applications [11,12,13,14]. 

Furthermore, decreased extracellular pH activates proton-sensitive receptors, such as certain G protein-coupled receptors (GPCRs), acid-sensitive ion channels (ASICs), transient receptor potential vanilloid channels (TRPVs), TWIK-related acid-sensitive potassium channels (TASKs) and transient receptor-gated channels (TRPCs). We recently published first data on the expression profiles of pH-GPCRs, TASKs/TRPVs and ASICs in various skin tumors [5,15,16,17]. 

TRPC4 in particular is regarded as a proton-sensitive, non-selective, receptor-operated cation channel which is important for calcium homeostasis, and its role in cancer may involve changes in the intracellular Ca^2+^ concentration [18,19]. An increase in intracellular Ca^2+^ concentration has been known to play a crucial role in angiogenesis and arterial remodeling [20]. Dysregulation of TRPC4 may interrupt Ca^2+^ homeostasis in cancer cells, which may activate signaling pathways that are highly associated with cancer progression, especially cancer chemoresistance. Calcium dysregulation might serve as a marker for melanoma prediction and influence the melanoma microenvironment, including immune cells, the vascular network, and chemical and physical surroundings [21]. TRPC4 was found to be closely associated with incidence of head and neck cancer and poor survival of patients with kidney cancer [22]. Nevertheless, there is no sufficient information about the presence and function of TRPC4 in skin tumors, but all these data suggest that Ca^2+^ influxes mediated by TRPC4 may contribute significantly to epidermal keratinocyte pathophysiology [23]. In epidermal cells, an increase in intracellular calcium concentration is an important event that triggers their differentiation [24,25]. It is known that an increase in extracellular calcium concentration above 0.1 mM, i.e., a “calcium switch”, triggers keratinocyte differentiation by mechanisms related to the increase in [Ca^2+^] that are not fully understood [25,26]. In this study, we investigate for the first time the expressions of proton-sensitive TRPC4 in basal cell carcinoma (BCC), squamous cell carcinoma (SCC), malignant melanoma (MM) and in nevus cell nevi (NCN).

## 2. Results

### 2.1. Controls

We established the immunohistochemical staining for TRPC4 on skin. Negative controls with isoantibodies and w/o primary antibodies were carried out (see Methods below). Testis and placenta served as positive controls (Figure 1a, expression in other organs shown in Figure 1b).

### 2.2. Samples

To determine the expression level of TRPC4, we initially collected 57 samples for BCC, 44 samples for SCC, 19 samples for NCN and 16 samples for MM. Due to processing errors, such as overstaining or very weak staining in the histological specimens, fewer samples were suitable for evaluation (see methods). Additionally, some samples did not contain tumor formations or tumor cell nests, which again reduced the amount of assessable samples as follows: BCC n = 39, SCC n = 27, NCN n = 13, MM n = 13. A panel with representative stainings is given in Figure 2. A detailed list of the staining and scoring results can be found in Appendix A. 

### 2.3. BCC

For BCC, nodular, sclerosing and mixed forms (partially nodular and sclerosing), as well as superficial forms, were evaluated (twenty-nine out of thirty-nine nodular, three out of thirty-nine superficial, six out of thirty-nine mixed, one out of thirty-nine sclerosing). In 51.3%, a negative reaction for TRPC4 was detected. A total of sixteen out of thirty-nine (41%) tissue samples were classified as weak positive or displayed a partial positive reaction. Only three out of thirty-nine (7.7%) samples showed a positive reaction, with >80% of cells being positive for TRPC4 and therefore classified as strong positive (Figure 3, Appendix A).

### 2.4. SCC

All 25 SCC samples displayed a weak positive expression for TRPC4 (Figure 3, Appendix A).

### 2.5. NCN

For NCN, evaluation of the samples was divided into epidermal and dermal portions. Ten out of thirteen tissue samples exhibited weak positive staining in both the epidermal and dermal sections. One out of thirteen appeared to be negative for TRPC4 in the epidermal as well as the dermal part. Only one tissue sample showed strong expression in the epidermal layer and a weak positive reaction for the dermal tissue area. Two out of thirteen histological specimens showed a decreasing expression towards deeper tissue levels. To conclude, for the epidermal section 15.4% appeared to be negative, 76.9% showed a weak positive reaction and 7.7% displayed a strong positive reaction. Evaluating the dermal parts, on the other hand, again 15.4% showed a negative reaction and 84.6% displayed a weak positive reaction. No strong positive staining could be detected in the dermal section for NCN (Figure 3, Appendix A). 

### 2.6. MM

The majority (92.3%) out of the thirteen samples of MM showed a weak positive staining in the epidermal and dermal section. The remaining histological specimen revealed a strong positive reaction in the epidermal portion and a weak positive reaction in the dermal areas (Figure 3, Appendix A). 

### 2.7. Statistical Analysis of TRPC4—Comparison of All Entities

TRPC4 expression was significantly lower in BCCs when compared to SCCs (*p* < 0.001 Kruskal–Wallis and post hoc Bonferroni tests), as well as between BCCs and epidermal (*p* < 0.015 Kruskal–Wallis and post hoc Bonferroni tests), and dermal parts of NCN (*p* < 0.046 Kruskal–Wallis and post hoc Bonferroni tests), respectively. The BCC expression of TRPC4 was also significantly lower in BCC than in epidermal (*p* < 0.001 Kruskal–Wallis and post hoc Bonferroni tests) and dermal parts of MMs (*p* < 0.012 Kruskal–Wallis and post hoc Bonferroni tests). No significant differences in TRPC4 expression were found between NCN and MM, NCN and SCC or SCC and MM (Table 1 and Appendix A). 

## 3. Discussion

We examined the expression profiles of TRPC4 in the most common types of skin cancer. There is a high probability of channel/receptor interplay between pH sensitive GPCRs, TASKs, and TRPCs [27,28]. One of the most striking results found was the lower frequency of TRPC4 expression in BCC when compared to SCC, NCN and MM. Protein expression was obviously altered between BCCs and SCCs, as seen in our immunohistochemistry stainings. BCCs seemed to not express pH-sensitive TRPC4 in half of the cases investigated in this study.

Studies on cBioportal were analyzed for mutation frequencies of TRPC4 in the investigated tumor entities. For MMs, up to 14% mutation frequency was found for TRPC4 (Appendix A). For non-melanoma skin cancer, SCCs showed in one study a markedly higher and in another study the same amount of mutation frequency than BCCs (Appendix A). However, there is no difference in the RNA expression levels of the investigated proteins according to studies analyzed at NCBI Geo (Appendix A).

Protein expression in our study was found to be reduced in BCCs as compared to SCCs, which could be due to post-transcriptional processes. 

TRPC4 is evenly expressed in SCCs, MMs and in NCN in both epidermal and dermal portions. In contrast to these tumor entities, BCCs were negative in ~50% of the samples, which is a striking immunohistochemical feature and similar to the results we found for GPR31, GPR151, TASK1, TASK3 and ASIC2 [5,15]. Even though there is little information on the expression of TRPC4 in skin tumors, the role in cancer progression has been proven in other tissues and it may represent potentially attractive targets for cancer therapeutics [29]. A substance called englerin A, isolated from the African plant phyllanthus engleri, can activate the ion channels TRPC4 and TRPC5 at nanomolar concentrations and it can inhibit the growth of tumor cell lines expressing high levels of TRPC4 or TRPC5. However, englerin A is extremely toxic in rodents, although it is very unstable in rodent serum [30,31]. The effect of an agent called tonantzitlolone (TZL) is at least superficially similar to that of englerin A [27]. Nevertheless, TZL is chemically distinct from englerin A and could therefore present important new opportunities because the unacceptable in vivo toxicity of englerin A in healthy rodents has been identified as a potential barrier to its development towards therapeutics [30]. TRPC4 was found to be closely associated with incidence of head and neck cancer and poor survival of patients with kidney cancer [22]. The overexpression of TRPC4 increased ovarian cancer cell colony growth [28]. Ca^2+^ influxes mediated by TRPC4 may contribute significantly to epidermal keratinocyte pathophysiology [23]. In epidermal cells, an increase in intracellular calcium concentration is an important event that triggers their differentiation [24,25]. It is known that an increase in extracellular calcium concentration above 0.1 mM, i.e., a “calcium switch”, triggers keratinocyte differentiation by mechanisms related to the increase in [Ca^2+^] that are not fully understood [25,26]. Taking these considerations and our results into account together, TRPC4 might serve as a potential diagnostic target in skin cancer, but more functional studies are required to fully understand the role of TRPC4 in skin tumor progression.

## 4. Materials and Methods

For all experiments, paraffin-embedded tissue samples from the dermatopathological Department of Dermatology, University Medical Center, Regensburg were used.

All tissue samples were older than 10 years, and therefore free to use under German legislation. Assessed samples were obtained from affected areas of patients with localized skin tumors. 

### 4.1. Immunohistochemistry

Using a microtome, tissue samples (embedded and fixed in paraffin) were cut into 3 µm-thick sections and afterwards fixated on slides. Hematoxylin and eosin staining was performed on each slide. This and all other following staining steps were conducted at room temperature. To guarantee comparability and minimize differences in staining of the four different tumor subtypes, each antibody staining step on every tumor entity was performed within two days.

Paraffin was removed from the tissue sections by incubating them for 60 min at 72 °C.

Afterwards, the slides were rehydrated with decreasing alcohol concentrations following the given protocol: 3 × xylol for 10 min, 2 × 100% ethanol for 5 min, 2 × 96% ethanol for 5 min, 2 × 70% ethanol for 5 min. To avoid false-positive results, endogenous peroxidase was blocked with 3% H_2_O_2_ (Fisher Scientific, No. 1404697) for 10 min. Simultaneously, an acidic citrate buffer with pH 6 (Zytomed, Bargteheide, Germany, REF ZUC028) was boiled for 30 min.

The slides were washed in distillated water and then boiled for 20 min in the prepared precooked citrate buffer, and then cooled for 20 min. Afterwards, they were transferred to Phosphate Buffer Solution (PBS) (Sigma-Aldrich, Darmstadt, Germany, No. D8537) for 10 min, followed by their fixation on cover slides and additional washing with PBS. To avoid unspecific antibody binding, proteins were blocked with blocking solution (ZytoChem Plus HRP Kit/Rabbit, Zytomed, Bargteheide, Germany, REF HRP060-Rb) for 10 min. Tissue sections were incubated with the primary goat anti-human TRPC4 (4.75 µg/mL Anti-TRPC4 antibody RRID: AB_10980235) polyclonal antibodies at 4 °C overnight.

The very next day, the slides underwent washing with PBS in three consecutive sessions. The tissue sections were then incubated with the secondary biotinylated antibody for 30 min, they were washed again three times with PBS, then incubated with streptavidin-HRP-conjugate for 20 min and washed 3× with PBS. Positive controls were stained with AEC plus (Dako, Santa Clara, CA, USA, No. K 3469) until the requested staining appeared. This took up to 6 min for SCCs, 6 min for BCCs, 8 min for NCN, and 5 min for MMs. The reaction was stopped with distillated water, and positive controls were counterstained with Mayer’s Haemalm (Roth, Karlsruhe, Germany, No. T865.3). The slides were scanned with PreciPoint M8, and the digital images were edited with ViewPoint online (PreciPoint, Freising, Bavaria, Germany).

### 4.2. Western Blot 

The signal of the above-mentioned primary goat anti-human TRPC4 was confirmed via Western blot (Appendix A). Proteins were separated using a 10% Mini-Protean^®^ TGX precast gel (BioRad, Hercules, CA, USA). After blotting, PVDF membranes were blocked with 5% skimmed milk in TBS buffer with 0.1% Tween for 1 h at room temperature and incubated with the primary antibody overnight. Incubation with the secondary antibody was performed for 1 h at room temperature. Blots were analyzed using the chemiluminescence system Fusion Pulse 6 (Vilber Lourmat, Collégien, France). For the detection of multiple antigens, antibodies were removed by incubation with the re-blot mild stripping solution (Merck, Rahway, NJ, USA) for 15 min. Primary antibodies: goat anti-TRCP4 (Thermo Fisher Scientific, Waltham, MA, USA), rabbit anti-actin (Sigma-Aldrich); secondary antibodies: rabbit anti-goat, goat anti-rabbit (both Agilent/Dako, Santa Clara, CA, USA).

### 4.3. Scoring

Experienced dermatopathologists from the Department of Dermatology and the Institute of Pathology at the University Medical Center, Regensburg evaluated the stainings visually. The epidermis was chosen as a reference structure to assess staining intensity. Sections were classified as ++ for strong positive reactions (>80% of cells either being positive or showing high intensity staining), + for weak positive/partial positive reaction (20–80% of cells displaying positive and staining weak or partially strong), -negative reaction (<20% of cells showing weak staining). Inconsistent staining throughout the tumor samples, as well as decreasing expression levels towards deeper tissue layers, were classified as weak positive. 

### 4.4. Statistics

First, ratings for all entities were compared using Kruskal–Wallis tests. For NCN and MMs, epidermal and dermal portions were separately used for testing. Pairwise comparisons were made via Bonferroni tests. Secondly, pairwise comparisons of BCCs vs. SCCs and of MMs vs. NCN were made for each protein using a Mann–Whitney U test, and the results are given as exact significance (shown as 2*(1-tailed significance), not corrected for ties, for BCCs vs. SCCs and epidermal portions of NCN/MMs) or asymptotic significance (2-tailed, for dermal portions of NCN/MMs).

## 5. Conclusions

First of all, our findings need to be confirmed in a larger sample size and by the study of different patients, especially the BCC growth patterns, to investigate the different roles of TRPC4 more precisely. In terms of methodology, the automated evaluation of immunohistochemistry could be a potential approach for future studies. Cell lines with knockout and overexpression of pH-sensitive TRPC4 could be subjected to varying pH_e_ in order to examine the role of this protein in proliferation, migration and cell survival. After identifying individual protein levels in different cell types (qPCR, Western blot), the next step is to use knockdown (siRNA)/knockout (CRISPR/Cas9) and overexpression strategies in combination with functional cellular assays to answer these questions, as we proposed before [16].

As this is the first study on the expression of the pH-sensitive protein TRPC4 in skin tumors in the literature, our approach is descriptive. However, one of the most striking results found was the lower frequency of TRPC4 expression in BCC when compared to SCC and melanocytic tumors (NCN/MM). The same result was seen in our previous studies for ASIC2 and GPR31, especially with regard to low expression in BCC and 100% expression in SCC [5,15]. This makes the study of TRPC4 a novel histological/diagnostic tool to distinguish these two entities (BCC/SCC). 

As a further step, it would then be important to conduct clinical studies on whether these receptors/channels have a significant effect on the overall survival of MM patients [32], because mutation frequencies of TRPC4 are quite high. 

## Figures and Tables

**Figure 1 ijms-24-01037-f001:**
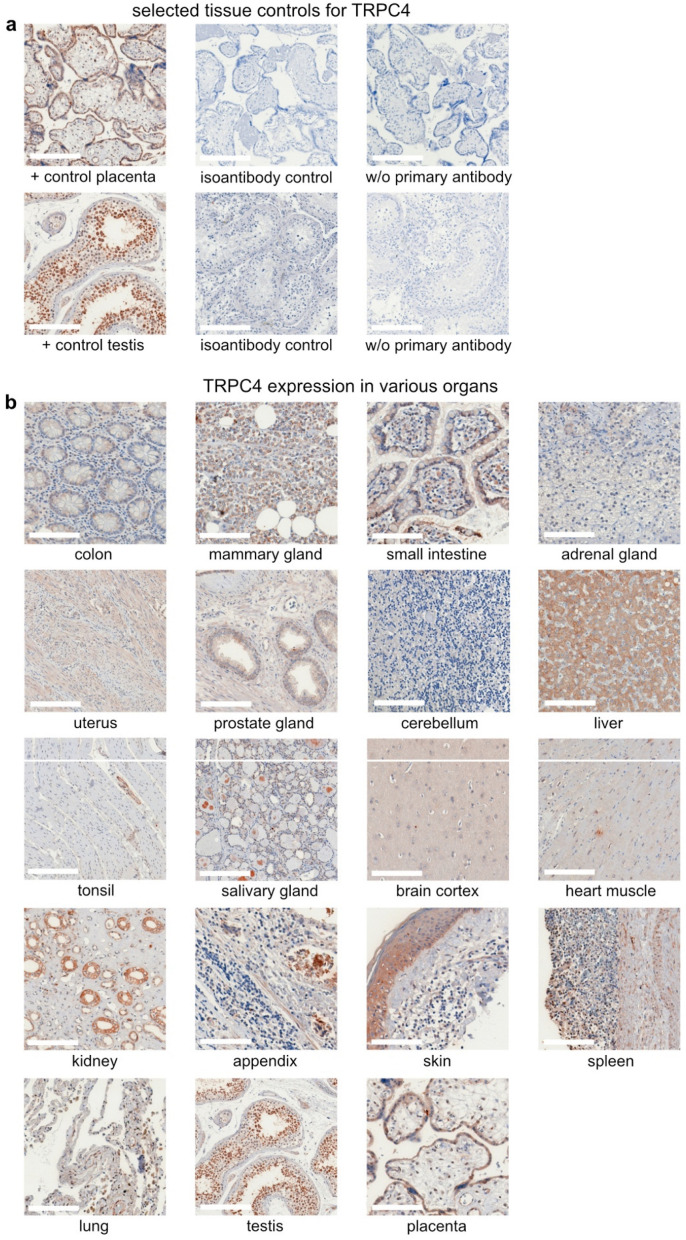
Tissue Controls for the Immunohistochemical staining of TRPC4. Scale bars represent 200 µm. (**a**) placenta und testis were used as positive controls for TRPC4 and showed no staining in relation to isoantibody control and without primary antibody. (**b**) TRPC4 expression in various organs.

**Figure 2 ijms-24-01037-f002:**
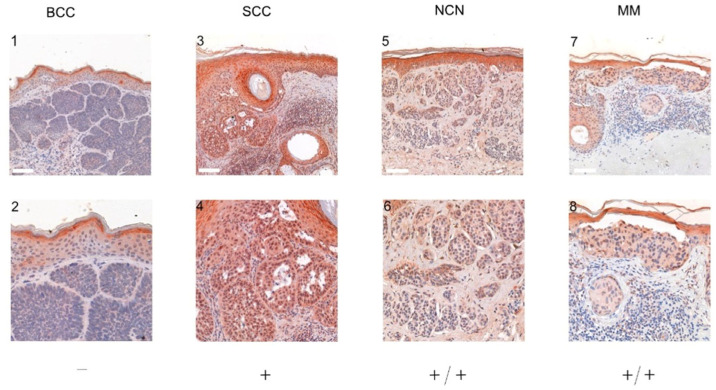
Representative immunohistochemical staining for TRPC4 in BCC, SCC, NCN and MM tissue. The tumor cells of BCC (example shown in **1**–**2**: patient 9 = slide number A1M_11S_1, see Appendix A) show very low expression of TRPC4 (all additional BCC cases in Appendix A). The SCC (example shown in **3**–**4**: patient 11 = slide number A4M_16S_1, see Appendix A) shows positive expression of TRPC4 (all additional SCC cases in Appendix A). The dermal and epidermal portions of NCN (example shown in **5**–**6**: patient 13 = slide number K. NZN TRPC4 9629-09, see Appendix A) appear weak positive for TRPC4 (all additional NCN cases in Appendix A). The epidermal and dermal sections of MM (example shown in **7**–**8**: patient 9 = slide number MM TRPC4 7060-09) show a weak positive expression of TRPC4 (additional cases Appendix A). For more stainings of other NCN, MM, SCC and BCC, see Appendix A. Scale bars represent 200 µm.

**Figure 3 ijms-24-01037-f003:**
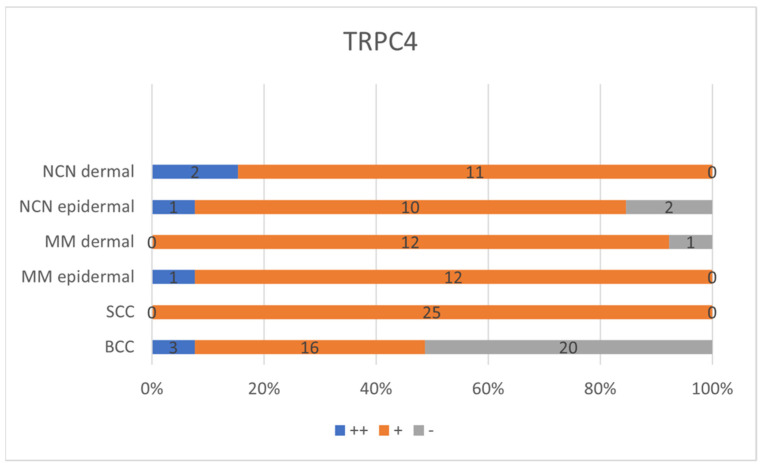
Summary of immunohistochemical scoring results for TRCP4 on NCN, MMs, SCCs and BCCs. ++/blue bar: strongly positive staining with >80% of cells positive and/or staining intensity is high; +/orange bar: 20–80% of cells show a weakly positive/partially positive reaction; −/grey bar: <20% of cells with weak staining (=negative reaction). NCN and MM are subdivided into epidermal and dermal portions. Numbers in bars represent the occurrence of the particular score. For additional information on the individual scores, see Appendix A.

**Table 1 ijms-24-01037-t001:** Statistical analysis of TRPC4—comparison of all entities.

Pairs	*p*-Value	Adjusted *p*-Value
BCC–NCN dermal	0.046	0.686
BCC–NCN epidermal	0.015	0.230
BCC–MM dermal	0.012	0.177
BCC–SCC	0.000	0.002
BCC–MM epidermal	0.001	0.008
NCN dermal–NCN epidermal	0.728	1.000
NCN dermal–MM dermal	0.672	1.000
NCN dermal–SCC	0.331	1.000
NCN dermal–MM epidermal	0.232	1.000
NCN epidermal–MM dermal	0.939	1.000
NCN epidermal–SCC	0.566	1.000
NCN epidermal–MM epidermal	0.396	1.000
MM dermal–SCC	0.627	1.000
MM dermal–MM epidermal	0.440	1.000
SCC–MM epidermal	0.690	1.000

Results of testing with Kruskal–Wallis test and post hoc Bonferroni comparison.

## Data Availability

Available upon reasonable request.

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
