# Peer review of "Expression of pH-Sensitive TRPC4 in Common Skin Tumors"

_ijms, 2023, doi:10.3390/ijms24021037_

Round 1
Reviewer 1 Report
We congratulate the authors for the subject approached in the manuscript under evaluation. As can be seen, although it is a pioneering study, the results are relevant to highlight the major role that TRPC4 plays in skin cancer.
In order to improve the quality of the manuscript, we recommend the following revisions to the authors:
- for the entire manuscript – in vivo (use italics) (eg, lines 58, 246);
- for the entire manuscript – use super or subscript if needed Ca2+, H2O2;
- first entry of the term within the manuscript – description for PBS abbreviation – Phosphate Buffer Solution (line 107);
- proper form for subchapter (line 149);
- proper form – skin cancer (line 283 and Supplementary Figure S10)
- replace "need to be verified by" with "need to be confirmed on" (line 260).
Reviewer 2 Report
The paper describes the expression profiles within different skin tumour entities based on a single antibody staining.
The specificity of the obtained signal is not proven by e.g. WB or in situ RT-PCR experiments.
Their is no rationale for the scoring system.
Why was no densitometric analysis applied?
In case of overstaining, why was the IHC not repeated?
Round 2
Reviewer 2 Report
No further comments